# Effect of girls' education on cancer awareness and screening in a natural experiment in Lesotho

Janny Liao [1,2] ✉, Ramaele Moshoeshoe [3,4], Michelle D. Holmes [5,6], S. V. Subramanian [7,8] & Jan-Walter De Neve [1,9]

Breast and cervical cancers are important causes of disability and premature death among women in Sub-Saharan Africa. Previous research has linked girls' education to cancer service access. Here, we examine the causal effect of girls' educational attainment on cancer screening practices by means of a natural experiment in Lesotho. In particular, we exploit variation in educational attainment among women that was introduced by an educational policy (a school-entry age cut-off). Data on awareness towards breast cancer, knowledge of Pap smear, breast self-exam, breast clinical exam, and having received a Pap smear is extracted from the Lesotho Demographic and Health Surveys 2009-10 and 2014 ($N = 7971$). Each additional year of schooling caused by the education policy increases awareness of breast cancer by 4.7 percentage points ($p = 0.014$, 95% Confidence Interval [CI]: 1.0, 8.5), awareness of Pap smear by 5.9 percentage points ($p = 0.001$, 95% CI: 2.3, 9.5), and engagement in Pap smear by 3.5 percentage points ($p = 0.004$, 95% CI: 1.1, 5.8). We found no statistically significant effects on breast self-exam and breast clinical exam.

Globally, cervical and breast cancer are important causes of cancer-related disability and premature death for women[1,2]. The incidence of breast cancer and cervical cancer exceeded 100,000 cases each in Sub-Saharan Africa (SSA) in 2021. Moreover, in Lesotho, the death rate for cervical cancer is three times as high as the rate in SSA, and the death rate for breast cancer is 1.6 times as high as SSA. Late-stage diagnosis of cancer in low-and middle-income countries (LMICs) is common and delay in treatment is further exaggerated by the limited access to comprehensive cancer treatment centers[3]. While mammography is the most effective screening method, resources can be highly constrained in LMICs, which makes the most efficient diagnostic methods

often clinical breast examination (CBE) or self-breast examination (SBE)[4]. Late diagnosis is largely attributed to the lack of awareness toward breast cancer[3]. Despite numerous campaigns promoting breast cancer awareness, women's awareness in LMICs is limited[3]. For cervical cancer, Pap smear, while effective, is hard to deliver systematically in LMICs due to the lack of trained providers, overburdened health facilities, insufficient supplies, and inadequate lab infrastructure[5]. Thus, cervical cancer rates are high despite efforts to implement national screening and treatment programs[5]. Currently, in Lesotho, there are few comprehensive cervical cancer screening programs established on a national scale, and there is a lack of advanced staged cancer

[1]Heidelberg Institute of Global Health, Faculty of Medicine and University Hospital, University of Heidelberg, Heidelberg, Germany. [2]Harvard College, Harvard University, Cambridge, MA, USA. [3]Department of Economics, National University of Lesotho, Roma, Lesotho. [4]Global Education Analytics Institute, Nairobi, Kenya. [5]Channing Division of Network Medicine, Department of Medicine, Brigham and Women's Hospital and Harvard Medical School, Boston, MA, USA. [6]Department of Epidemiology, Harvard T. H. Chan School of Public Health, Boston, MA, USA. [7]Harvard Center for Population and Development Studies, Harvard University, Cambridge, MA, USA. [8]Department of Social and Behavioral Sciences, Harvard T.H. Chan School of Public Health, Boston, MA, USA. [9]Division of Global Health Management and Policy, School of Public Health, San Diego State University, San Diego, CA, USA. ✉e-mail: jannyliao@college.harvard.edu

treatment facilities[6]. HPV vaccination for adolescent girls was recently reintroduced in Lesotho and achieved at least about 70% coverage in 2022[7].

Various factors have been hypothesized to be related to awareness of breast cancer and cervical cancer screening in LMICs, including educational attainment and household wealth[8,9]. A large literature has assessed the correlation between increased education and the awareness and use of cancer screening services. Additional formal education is generally positively associated with the probability of screening[10,11]. A past study identified a causal relationship of the effects of education on the probability of breast cancer incidence and survival for a high-income country[12]. However, most of these studies have yet to establish a causal impact of education on cancer screening practices for LMICs. The relationship between education and screening is likely confounded by other factors – such as socioeconomic status, family formation (e.g., marital status, number of children), as well as psychological and cultural traits—which may be related to both educational attainment and cancer outcomes[13,14]. Additionally, no randomized controlled trials have delivered a formal education intervention systematically and on a national scale. While there have been studies on the causal impacts of education on cancer incidence and mortalities in higher-income settings, to our knowledge, no studies have investigated the causal effect of girls' education on women's cancer awareness and screening practices in LMICs[15,16].

In the absence of randomized controlled trials (RCTs), natural experiments offer an important opportunity for causal inference in population health[17,18]. In Botswana, for example, a natural experiment investigated the effect of additional secondary schooling on HIV infection risk using variation in educational attainment induced by an education policy which reformed the grade structure of secondary school, expanding access to grade ten for affected birth cohorts[19]. Natural experiments can complement evidence from RCTs and have several strengths[20]. For example, to estimate a policy's "real-life" effectiveness as opposed to clinical efficacy or to evaluate interventions that have become standard practice without preceding RCTs[21]. Policy makers, for example, may be more concerned with de facto effects of large-scale programs as opposed to effects under ideal conditions such as during an RCT[22]. Additionally, natural experiments can be less costly because they can be implemented using existing data. Moreover, in studies focused on formal education, it would be ethically impermissible to randomly allocate individuals to receive or be deprived of educational opportunities.

In this work, we exploit an education policy which induces quasi-random variation in total years of schooling by children's month of birth in Lesotho. The basic framework underpinning causality in our study is similar to that of an RCT, where participants are assigned to a treatment and control group. In Lesotho, a school-entry age cut-off based on the month of birth induced more education among children who were born after the cut-off (treatment group) compared to children who were born before the cut-off (control group)[23]. Because parents have limited control over the exact timing of the month of birth of their children, we can compare the long-run outcomes of children as if they were randomized into treatment and control groups by the education policy[24]. To analyze this natural experiment, we first document the impact of the education policy on total years of schooling completed using nationally representative data. We then compare cancer outcomes among women who were affected by the education policy in childhood, controlling for year of birth and age effects as well as survey (period) effects[24,25]. We find that additional education caused by the education policy increases breast cancer and Pap smear awareness as well as engagement in Pap smear. Increasing girls' educational attainment could improve awareness and screening rates of breast and cervical cancer for women in Lesotho.

## Results
### Descriptive statistics
Table 1 shows the selected descriptive statistics of our analytical sample. The table displays the characteristics of 7971 eligible female respondents in the 2009–10 and 2014 DHS surveys. The average duration of schooling increased by 0.3 years from 2009–10 to 2014 and averages 7.8 years (range: 0, 18). For breast cancer-related screening practices, despite the high awareness levels toward breast cancer (88.9%), screening practices were low. Among women aged 25–49 years in the pooled sample (both survey years), 32.3% engaged in self-examination practices, and 7.8% of women had a clinical breast examination on an annual basis. Pap smear awareness levels increased by nearly 50% between 2009–10 and 2014 (from 37.2 to 55.1%). Screening practices for cervical cancer, however, were very low. Among women aged 25–49 years, 7.9% of women had performed a Pap smear in the past 12 months in 2009–10, and 14.0% of women had ever performed a Pap smear in 2014. Similar to breast cancer screening practices, awareness of the disease and preventive measures appear to be higher than actions to prevent cancer (e.g. clinical exam). Health insurance coverage was also uncommon and decreased considerably between 2009–10 and 2014. In the pooled sample, only 8.0% of women were covered by any health insurance.

### Association between girls' education and cancer awareness and screening
In Fig. 1, we graphically display the unadjusted relationship between educational attainment and women's cancer awareness and screening using pooled data from the 2009–10 and 2014 DHS surveys. For breast self-examination practices, breast clinical examination practices, awareness of Pap smear, and the probability of ever having a Pap smear, there is a positive gradient between increased educational attainment and women's cancer awareness and screening. These relationships persisted in multivariable OLS regression models controlling for indicators for the year of birth and survey year. In Table 2, we show regression results for the association between educational attainment and women's cancer awareness and screening, separately for each outcome. For example, in the pooled sample, each additional year of schooling among women is associated with a 2.9 percentage point increase ($p < 0.001$, 95% CI: 2.6, 3.2), 0.7 percentage point increase ($p < 0.001$, 95% CI: 0.5, 0.9), 5.3 percentage point increase ($p < 0.001$, 95% CI: 5.0, 5.6), and 2.2 percentage point increase ($p < 0.001$, 95% CI: 2.0, 2.5) in breast self-exam, breast clinical exam, Pap smear awareness, and the probability of having done a Pap smear, respectively. While suggestive, these regression results may be vulnerable to residual confounding. We, therefore, turn to a quasi-experimental approach to determine the causal effect of education on cancer awareness and screening in the next section.

### Exogenous instrument: educational attainment by month of birth in Lesotho
In Fig. 2, we illustrate the total years of schooling completed by the birth month of women aged 25 to 49. The binned scatter plot suggests about 0.4–0.7 years of more total schooling years for women born between July and December compared to women who were born between January and June. We further show that this relationship of increased total years of school for women born after the June 30th school-entry cut-off holds in multivariable OLS regression models shown in Table 3. The magnitude of the change in total schooling ranged from 0.5 ($p = 0.002$, 95% CI: 0.17, 0.75) to 0.6 additional years of schooling ($p < 0.001$, 95% CI: 0.47, 0.75) across different model specifications in our regression models. We observed the largest absolute differences toward the end of primary school (completed 7 years of schooling or more) and in junior secondary school (completed at least 8-10 years of schooling or more). For example, women born between July and December were 8.4 percentage points ($p < 0.001$,

**Table 1 | Selected characteristics of study respondents**

| Subsample: Women | DHS 2009–10 | DHS 2014 | Pooled (2009–2014) |
|---|---|---|---|
| Age, mean years (range) | 35.2 (25–49) | 35.0 (25–49) | 35.1 (25–49) |
| Total schooling completed, mean years (range) | 7.7 (0–18) | 8.0 (0–18) | 7.8 (0–18) |
| Marital status | | | |
| Ever married, N (%) | 3461 (82.2) | 3107 (82.6) | 6568 (82.4) |
| Never married, N (%) | 747 (17.8) | 656 (17.4) | 1403 (17.6) |
| Heard of breast cancer | | | |
| Ever heard of breast cancer, N (%) | n/a | 3345 (88.9) | 3345 (88.9) |
| Never heard of breast cancer, N (%) | n/a | 418 (11.1) | 418 (11.1) |
| Breast self-exam | | | |
| Performed in last 12 months, N (%) | 1111 (26.4) | 1463 (38.9) | 2574 (32.3) |
| Not performed in last 12 months, N (%) | 3097 (73.6) | 2300 (61.1) | 5397 (67.7) |
| Breast clinical exam | | | |
| Performed in last 12 months, N (%) | 222 (5.3) | 396 (10.5) | 618 (7.8) |
| Not performed in last 12 months, N (%) | 3986 (94.7) | 3367 (89.5) | 7353 (92.3) |
| Heard of Pap smear | | | |
| Ever heard of Pap smear, N (%) | 1567 (37.2) | 2075 (55.1) | 3642 (45.7) |
| Never heard of Pap smear, N (%) | 2641 (62.8) | 1688 (44.9) | 4329 (54.3) |
| Performed Pap smear | | | |
| Performed a Pap smear, N (%) | 334 (7.9) | 527 (14.0) | 861 (10.8) |
| Did not perform a Pap smear, N (%) | 3874 (92.1) | 3236 (86.0) | 7110 (89.2) |
| Health insurance coverage | | | |
| Covered by any health insurance, N (%) | 545 (13.0) | 90 (2.4) | 635 (8.0) |
| Not covered by health insurance, N (%) | 3663 (87.1) | 3673 (97.6) | 7336 (92.0) |
| Decision on respondent's healthcare | | | |
| Respondent alone, N (%) | 1311 (31.2) | 1085 (28.8) | 2396 (30.1) |
| Respondent and partner, N (%) | 1083 (25.7) | 1288 (34.2) | 2371 (29.8) |
| Partner alone, N (%) | 467 (11.1) | 225 (6.0) | 692 (8.7) |
| Someone else, N (%) | 18 (0.4) | 14 (0.4) | 32 (0.4) |
| Other, N (%) | 3 (0.1) | 2 (0.1) | 5 (0.1) |
| No partner, N (%) | 1326 (31.5) | 1149 (30.5) | 2475 (31.1) |
| Observations | 4208 | 3763 | 7971 |

Table 1 shows selected characteristics of women aged 25–49 years with complete data on breast and cervical cancer awareness and screening. Women with missing information for any of the five breast and cervical cancer awareness and screening questions were removed from the analytical sample (n = 36), yielding a final analytical sample of 7971 women. Unweighted. n/a not available. Source: data from Lesotho DHS 2009–10 and 2014.

95% CI: 6.5, 10.4) and 4.6 percentage points more likely (p < 0.001, 95% CI: 2.8, 6.8) to have completed either at least 7 years of schooling or at least 10 years of schooling, respectively, compared to women born between January and June (Table S1). These results are also consistent with prior research which found that the education policy led to an increase in educational attainment among late starters[26]. This "natural experiment" provides an opportunity to estimate the causal impact of additional education on cancer awareness and screening by comparing cancer outcomes of cohorts born after the school-entry age cut-off versus cohorts born before the cut-off.

### Causal effect of girls' education on cancer awareness and screening

**Intention-to-treat (ITT) results.** In Fig. S4 in the Appendix, we show the binned scatter plots for cancer awareness and screening outcomes by women's month of birth. Women born between July and December (after the school-entry age cut-off), had on average higher knowledge of cancer and were more commonly screened for cervical cancer compared to those women who were born between January and June. These results are generally consistent with the gradients observed in Fig. 1 between increased years of education and cancer knowledge and screening. For awareness of cervical cancer, for example, we similarly

observe a positive relationship between being born between July and December (after the school-entry age cut-off) and the probability of being aware of Pap smear. However, we find little evidence for a relationship between exposure to the education policy and breast cancer screening (Fig. S4). In Model 1 of Table 4, we show ITT regression results for the relationship between exposure to the education policy and cancer awareness and screening, controlling for indicators for the year of birth and survey year, using the pooled data from the 2009–10 and 2014 surveys. Our ITT results were consistent with our findings shown in Fig. S4. Exposure to the education policy was related to breast cancer awareness, cervical cancer awareness, and the probability of screening for cervical cancer. The relationship between exposure to the education policy and breast cancer screening, however, did not reach conventional benchmarks of statistical significance (i.e., p < 0.05).

**Instrumental variable (IV) results.** In Model 2 of Table 4, we show regression results using exposure to the education policy as an IV for the respondent's total length of schooling completed. In 2SLS models, we observe a positive relationship between additional educational attainment and cancer awareness and screening practices. Each additional year of schooling caused by the

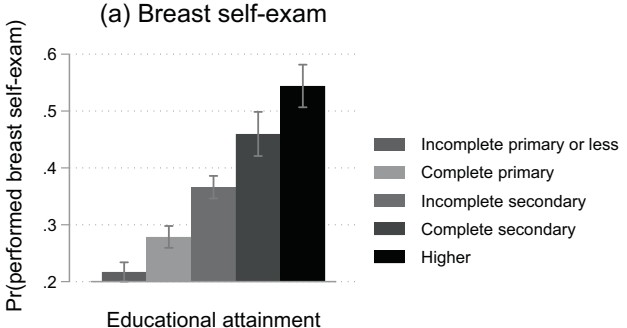

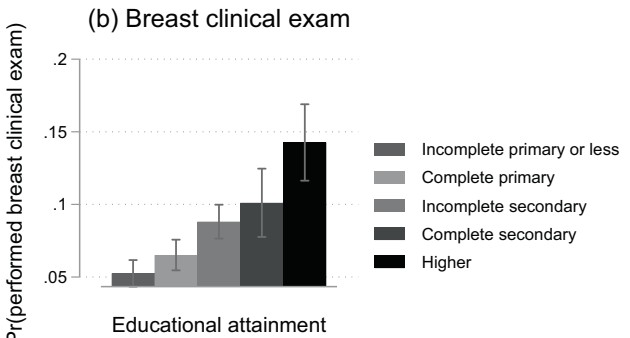

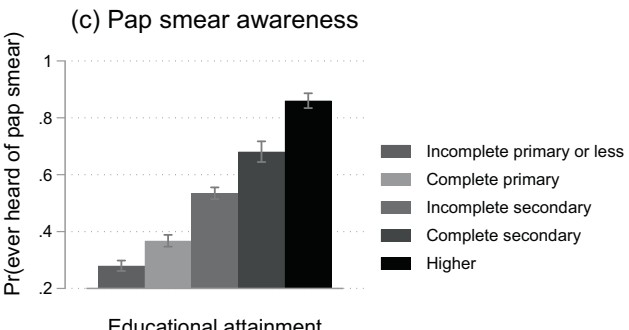

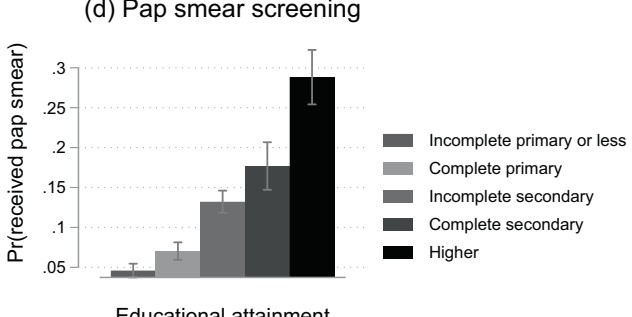

**Fig. 1 | Cancer awareness and screening by educational attainment in Lesotho.** Figure shows breast and cervical cancer awareness and screening by different educational attainment levels among women aged 25–49 years in Lesotho. Data were presented as the probability of each outcome and its 95% confidence intervals. Figure **a** shows the probability of having performed a breast self-exam (past 12 months), figure **b** shows the probability of having performed a breast clinical exam (past 12 months), figure **c** shows the probability of ever having heard of Pap smear, and figure **d** shows the probability of ever having received a Pap smear (past 12 months in the DHS 2009–10 and lifetime prevalence in the DHS 2014). Unweighted. Source: data from Lesotho DHS 2009–10 and 2014. *N* = 7971.

**Table 2 | Ordinary least squares regression results: the relationship between total years of schooling and women's cancer awareness and screening (N = 7971)**

| Dependent variable (DV) | Ever heard of breast cancer (1 = yes, 0 = no) | Breast self-exam (1 = yes, 0 = no) | Breast clinical exam (1 = yes, 0 = no) | Ever heard of Pap smear (1 = yes, 0 = no) | Performed Pap smear (1 = yes, 0 = no) |
|---|---|---|---|---|---|
| | Coef | Coef | Coef | Coef | Coef |
| Coefficient on years of schooling | 0.019 | 0.029 | 0.007 | 0.053 | 0.022 |
| 95% CI | [0.016, 0.022] | [0.026, 0.032] | [0.005, 0.009] | [0.050, 0.056] | [0.020, 0.025] |
| *p* value | <0.001 | <0.001 | <0.001 | <0.001 | <0.001 |
| Mean DV, January-June birth cohorts | 0.877 | 0.316 | 0.077 | 0.441 | 0.099 |
| R-squared | 0.046 | 0.060 | 0.020 | 0.155 | 0.075 |
| Observations | 3763 | 7971 | 7971 | 7971 | 7971 |

Table shows ordinary least regression (OLS) results for the relationship between additional schooling (years) and women's cancer awareness and screening practices. All models controlled for indicators for year of birth and survey year. We show 95% confidence intervals (CI) and two-tailed *p* values for all analyses. No multiple comparison adjustments were made. The sample included all women aged between 25–49 years in the Lesotho DHS 2009–10 and 2014. Data on ever heard of breast cancer was not available in the DHS 2009–10 survey. *N* = 7971.

education policy increased the probability of having heard of breast cancer, having heard of a Pap smear, and ever having done a Pap smear by 4.7 percentage points (*p* = 0.014, 95%CI: 1.0, 8.5), 5.9 percentage points (*p* = 0.001, 95%CI: 2.3, 9.5), and 3.5 percentage points (*p* = 0.004, 95%CI: 1.1, 5.8), respectively. The proportion of women who had performed a Pap smear among women born between January and June was 9.9%, implying a relative increase of more than one-third (35.4%) in engagement in Pap smear for each additional year of schooling. These estimates were larger in magnitude compared to our estimates from OLS models. One explanation for this finding could be that estimates obtained from 2SLS models are local to the subpopulation of

compliers—i.e., women who were induced to obtain more total years of schooling because of the education policy (Note S2).

**Results from supplementary analyses**

In Fig. S2 in the Appendix, we show a histogram of the month of birth among women to rule out seasonality in births and potential manipulation of birth dates. We observe no evidence of bunching in the density of birth dates at the June 30 threshold. In Tables S2–S7, we show results for each of our five cancer outcomes under different sample and model specifications. For example, when we include respondents between the ages of 15 to 49 years old (as opposed to ages 25 to 49 years old), each additional year of education corresponds with

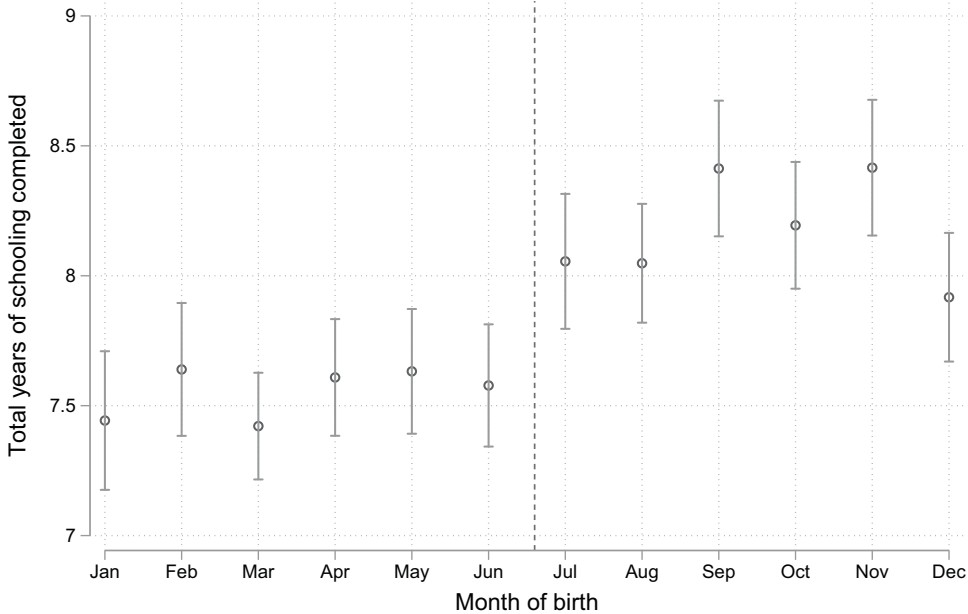

**Fig. 2 | Educational attainment by month of birth among women in Lesotho.** Figure shows years of total education against the month of birth among women aged 25–49 years. Data were presented as the mean of total years of schooling completed and its 95% confidence intervals. The dashed vertical line represents the school-entry age cut-off for primary school on June 30 in Lesotho. Unweighted. Source: data from Lesotho DHS 2009–10 and 2014. $N = 7971$.

**Table 3 | First-stage regression results: the relationship between being born after June 30 and total years of schooling completed among women in Lesotho ($N = 7971$)**

| Predictor: born between July and December (1 = yes, 0 = no) | Change in duration of schooling in years (Coef) | R-squared | Observations |
|---|---|---|---|
| Model specification | | | |
| 1: No control variables | 0.610 | 0.009 | 7971 |
| 95% CI | [0.470, 0.751] | | |
| p value | <0.001 | | |
| 2: Survey year | 0.607 | 0.016 | 7971 |
| 95% CI | [0.467, 0.747] | | |
| p value | <0.001 | | |
| 3: Survey year + YOB | 0.571 | 0.037 | 7971 |
| 95% CI | [0.432, 0.711] | | |
| p value | <0.001 | | |
| 4: Survey year + YOB + MOB | 0.489 | 0.037 | 7971 |
| 95% CI | [0.206, 0.772] | | |
| p value | <0.001 | | |
| 5: Survey year + YOB + MOB + Age | 0.463 | 0.040 | 7971 |
| 95% CI | [0.173, 0.754] | | |
| p value | 0.002 | | |

Table shows ordinary least regression (OLS) results for the relationship between being born after June 30 (binary) and total years of schooling completed. Model 1 includes no covariates; Model 2 includes indicators for the survey year; Model 3 includes indicators for the survey year and indicators for the year of birth (YOB); Model 4 includes indicators for the survey year, YOB, and a linear term in month of birth (MOB); Model 5 includes indicators for survey year, YOB, a linear term in MOB, and indicators for single-year age groups. The availability of two survey years allows us to generate variation in age for a given YOB so that age and YOB are not collinear. We show 95% confidence intervals (CI) and two-tailed p values for all analyses. No multiple comparisons adjustments were made. The sample includes all women ages 25–49 years in the Lesotho DHS 2009–10 and 2014 ($N = 7971$).

an increase of knowledge on Pap smear by 6.5 percentage points ($p = 0.005$, 95% CI: 1.9, 11.0). When the sample size is narrowed to ages 35 to 49 years old, each additional year of education corresponds with an increase in knowledge on Pap smear of 7.8 percentage points ($p = 0.001$, 95% CI: 3.2, 12.5). We also found qualitatively similar results when stratifying our sample by DHS survey year (period), restricting the sample of women to a narrower window of months of birth around the cutoff (±3 months as opposed to ±6 months) and when controlling for each woman's interviewer in addition to indicators for year of birth and indicators for survey year. Results for the relationship between additional education and breast cancer awareness among men are presented in the Appendix. The effect of male education on breast cancer awareness did not reach conventional benchmarks of statistical significance (i.e., $p < 0.05$) (Note S3 and Table S8).

## Discussion

Using an education policy as a natural experiment, we show that schooling may promote breast and cervical cancer screening by enhancing awareness and encouraging routine clinical screening practices in Lesotho. Specifically, our results demonstrated that additional years of schooling as a result of a school-entry age policy in Lesotho increases awareness of breast cancer, awareness of Pap smear, and screening with Pap smear. In terms of relative effect sizes, our results are comparable to known interventions to increase cervical screening practices[27]. In Nigeria, for example, a 6-month mHealth intervention with health promotion information increased the uptake of cervical cancer screening by 43% compared to standard of care (Risk ratio [RR]: 1.43, $p = 0.031$, 95% CI: 1.03, 1.98)[28]. Similarly, in our study, an additional year of schooling as a result of the education policy in Lesotho increased the likelihood of engagement in Pap smear by 43% (RR: 1.43, $p = 0.016$, 95% CI: 1.07, 1.90) (Table S7). Importantly, however, formal education has a myriad of additional benefits which are not taken into account in our analyses. Our estimates obtained from 2SLS models for breast cancer screening outcomes were smaller

**Table 4 | Intention-to-treat and two-stage least squares regression results: the relationship between girls' education and cancer awareness and screening**

| Dependent variable (DV) | Ever heard of breast cancer (1 = yes, 0 = no) | Breast self-exam (1 = yes, 0 = no) | Breast clinical exam (1 = yes, 0 = no) | Ever heard of Pap smear (1 = yes, 0 = no) | Performed Pap smear (1 = yes, 0 = no) |
|---|---|---|---|---|---|
| | Coef | Coef | Coef | Coef | Coef |
| Model 1: ITT models controlling for period and birth cohort fixed effects | | | | | |
| Predictor: born between July and December (1 = yes, 0 = no) | 0.026 | 0.009 | -0.001 | 0.034 | 0.020 |
| 95% CI | [0.006, 0.046] | [−0.011, 0.030] | [−0.013, 0.010] | [0.012, 0.055] | [0.006, 0.034] |
| $p$ value | 0.012 | 0.376 | 0.821 | 0.002 | 0.005 |
| $R$-squared | 0.008 | 0.022 | 0.014 | 0.044 | 0.025 |
| Observations | 3763 | 7971 | 7971 | 7971 | 7971 |
| Model 2: 2SLS models controlling for period and birth cohort fixed effects | | | | | |
| Coefficient on years of schooling | 0.047 | 0.016 | −0.002 | 0.059 | 0.035 |
| 95% CI | [0.010, 0.085] | [−0.019, 0.052] | [−0.023, 0.018] | [0.023, 0.095] | [0.011, 0.058] |
| $p$ value | 0.014 | 0.367 | 0.821 | 0.001 | 0.004 |
| Mean DV, January-June birth cohorts | 0.877 | 0.316 | 0.077 | 0.441 | 0.099 |
| F-statistic | 25.9 | 63.7 | 63.7 | 63.7 | 63.7 |
| Observations | 3763 | 7971 | 7971 | 7971 | 7971 |

Table shows intention-to-treat (ITT) and two-stage least squares regression results (2SLS) for the effect of girls' education on women's cancer awareness and screening. Model 1 is an ordinary least squares (OLS) linear probability model controlling for potential confounders. Model 2 is a two-stage least squares linear probability model in which exposure to increased schooling from the school-entry age policy was used as an instrumental variable for the respondent's duration of schooling (in years). All models controlled for indicators for the year of birth and survey year. We show 95% confidence intervals (CI) and two-tailed $p$ values for all analyses. No multiple comparison adjustments were made. Data on breast cancer awareness was not available in the DHS 2009–10 survey. The sample included all women aged between 25–49 years in the Lesotho DHS 2009–10 and 2014 ($N$ = 7971).

compared to those from more conventional OLS models, suggesting that estimates from OLS models for breast cancer screening may be vulnerable to residual confounding. Our estimates from 2SLS models are not vulnerable to unobserved factors (such as unmeasured socio-economic status or psychological traits), which may confound the association between increased educational attainment and breast cancer screening outcomes. These results were robust to a wide range of alternative specifications, including different specifications of our analytical sample, additional control variables, and a placebo test.

While there have been school-based programs that provide information on cervical cancer prevention, there was limited information on cancer and/or screening in the formal school curriculum in Lesotho at the time when most of the women included in our sample went to school. Lesotho's national HPV program was only recently reintroduced in 2022; it previously was developed in 2009, and ended in 2015 due to financial constraints[7,29]. Moreover, our results were qualitatively similar when restricting our sample to older cohorts who were not exposed to recent school-based HPV programs. One alternative explanation for the observed causal effect on cancer knowledge may be changes in skills development during middle childhood and adolescence[30]. School-entry age has large effects on reading skills among children in Lesotho, including on measured reading skills in English (literacy), as evidenced by late starters scoring higher on English reading assessments and reading more frequently (Fig. S5 and Table S10). This divergence in reading skills in childhood may have had large long-term consequences in adulthood. In Lesotho, where information on cancer prevention for adults is predominantly offered in a non-indigenous language (English), rather than Sesotho, women with a restricted understanding of English would have a lower probability of retaining the information[31]. Additionally, respondents born after the cut-off lived in wealthier households and in areas with improved access to healthcare services[32]. These alternative hypotheses may suggest how the causal mechanism occurs without direct exposure to screening-related programs.

These results have several policy implications. The average educational attainment in Lesotho is low (8 average years of schooling in the DHS 2014), suggesting that continuing to improve educational attainment in Lesotho may have further positive effects on cancer outcomes. Since LMICs have largely achieved universal primary education, policy efforts have increasingly focused their attention on increasing access to secondary education, especially among young women in LMICs. The UN Sustainable Development Goal #4, for example, calls for universal primary and secondary education by the year 2030[33]. Similarly, the Education Plus initiative, a joint initiative of UNAIDS, UNESCO, UNFPA, UNICEF, and UN Women, is a high-level political advocacy drive to accelerate investments in sub-Saharan Africa to prevent HIV infection with secondary education as the strategic entry point. Our results suggest that these efforts to increase girls' education may have knock-on effects on cancer awareness and screening. Estimates of the returns to investments in the length of educational attainment, which do not take into account these effects, will likely underestimate the larger societal benefits of investments in girls' education in Lesotho.

This study has some limitations. Firstly, in terms of GDP per capita, Lesotho is at the lower end among World Bank designated lower-middle-income countries[34]. Our study may not generalize to other settings. However, our data includes several nationally representative datasets from Lesotho, which cover a larger diversity of participants than those typically engaged in RCTs[18,21]. Our methodological approach also allowed for a relatively long follow-up period, from an exposure determined in early childhood to long-term health outcomes up until middle adulthood. This extended duration provides a substantial depth of data regarding the longer-term effects of education on health behaviors within the covered age range. Second, while our data includes participants up to the age of 45 years old, they may not be generalizable to older age groups. Older age groups could exhibit different educational impacts on health behaviors. Third, respondents may be influenced by social desirability bias when answering questions related to medical screening. The outcomes in this study were survey-based and no official medical records or biomarkers were included as part of this study. Our results were consistent, however, when assessing outcomes which were more easily

verified, such as measured literacy, household wealth, and distance between the household and the nearest clinic (Fig. S5 and Table S10). Fourth, sample sizes when restricting the sample of women to a narrower window of months of birth around the cutoff (±3 months as opposed to ±6 months) are smaller, and our results do not all remain as statistically strong (Tables S2–S6).

Increasing educational attainment could play a considerable role in improving breast and cervical cancer awareness, as well as Pap smear screening practices in Lesotho. Our findings support the relationship between education and cancer awareness and screening outcomes for women in this context. While extending women's access to formal education is essential for numerous developmental goals—such as improving critical thinking, problem-solving skills, and economic empowerment—this study highlights a potential additional benefit: increased cancer knowledge and cancer screening rates. However, additional empirical evidence is needed to determine whether improvements in educational attainment translate to reduced disability and premature deaths, lead to similar outcomes in the awareness and screening of other types of cancers, such as colorectal or lung cancer, and extend to broader health domains, such as chronic disease management and preventive care practices. Future studies could build on this natural experiment to better understand the full scope of the impact of girls' education on women's health in Lesotho.

## Methods

### Ethics and inclusion
The study was approved by the Heidelberg University Hospital Ethics Committee (S-825/2022). This research did not undergo data collection locally but has a local researcher in the authors' team. We have also taken local and regional research relevant to our study into account in citations.

### Data source
We used data from the 2009–10 and 2014 Lesotho Demographic and Health Surveys (DHS), cross-sectional, individual-level surveys with breast and cervical cancer-related knowledge and screening practices. Briefly, the DHS survey is a two-stage cluster randomized nationally representative survey implemented with support from USAID. The DHS surveys were fielded from October 2009–January 2010 (DHS 2009–10) and September–December 2014 (DHS 2014). All women aged 15–49 who were either permanent residents of the selected households or visitors who stayed in the household the night before the survey were eligible to be interviewed. The woman's questionnaire included questions on demographics, education, and breast and cervical cancer outcomes. Additionally, in half of the households, all men aged 15–59 years who were either permanent residents of the selected households or visitors who stayed in the household the night before the survey were eligible to be interviewed. Household response rates in the DHS were 98% (2009–10) and 99% (2014). For the 2009–10 survey, individual response rates were 98% among women and 95% among men. For the 2014 survey, individual response rates were 97% among women and 94% among men[35,36]. Cancer-related outcomes were not available in the Lesotho DHS of 2004-05 and among men in the DHS 2009–10 and, therefore, were not included in our analysis. Additional details on the DHS surveys are available elsewhere[37].

### Study population
The sample included all respondents ages 25 to 49 (women) and respondents ages 25 to 59 (men). We included respondents above the age of 24 so that they had the opportunity to complete their formal schooling. In sensitivity analyses, however, we also used alternative specifications of our analytical sample, including either women aged 15–49 years or women aged 35–49 years. Information on age, birth month, total years of education, and cancer-related outcomes was available for 99% of respondents, yielding a final sample of 4208

female respondents in the 2009–10 DHS, 3763 female respondents in the 2014 DHS, and 1707 male respondents in the 2014 DHS. Figure S1 in the Appendix displays a study participant flow diagram.

### Exposure
The key exposure in our analysis was the total years of schooling completed by the time of the survey. We assessed male and female schooling separately.

### Exogenous instrument
To identify causal effects, we leveraged the variation in educational achievement attributed to the Lesotho school entry eligibility policy. Lesotho's school year starts in January. Lesotho school entry eligibility policy stipulates that children must reach six years of age by June 30 to qualify for enrollment in public schools[38]. Children who will turn six by June 30 are eligible to start grade 1 in January of that year, whereas children who will turn six on July 1 or later will be eligible to start grade 1 in January of the following year. Consequently, each cohort in grade one exhibits up to a one-year age gap between the oldest and youngest students. Previous research has documented adverse outcomes associated with an earlier school start in LMICs[39]. In Lesotho, early starters are more likely to drop out of school and complete substantially fewer total years of schooling compared to late starters[26]. Late school entrants who are relatively old for a grade may be more mature, for example, compared to their younger classmates despite no differences in innate ability[40]. The school-entry age policy provides an exogenous instrument to determine the causal impact of total years of schooling on cancer-related screening outcomes in adulthood by comparing respondents who entered the school before and past the June 30 cutoff[17,18]. To do so, we defined the instrument as a binary indicator with the value of 1 if the respondent was born after the eligibility cut-off, and "not exposed" (=0) if the respondent was born before the eligibility cut-off for school entry[19]. Individuals born after June 30 in Lesotho are, on average, more educated and were classified as "exposed" in our model. Additional details on the education context and school-entry age policy are available in Note S1.

### Control variables
We controlled for indicators for year of birth and indicators for survey year to take into account survey (period) effects. The availability of two survey years allows us to generate variation in age for a given year of birth. By simultaneously controlling for year of birth and survey year, we are therefore also implicitly controlling for age effects (age and year of birth are not collinear with two surveys). In sensitivity analyses, we also considered a model without covariates and a model when including additional covariates, such as a continuous linear term in month of birth and interviewer fixed effects, to control for systematic differences in measurements across interviewers.

### Outcomes
Our main outcomes of interest were binary indicators for breast cancer awareness (ever heard of breast cancer), breast cancer self-screening (past 12 months), breast cancer clinical screening (past 12 months), cervical cancer screening awareness (ever heard of Pap smear), and cervical cancer clinical screening (past 12 months in the DHS 2009–10 and lifetime prevalence in the DHS 2014)[35,36].

### Statistical analyses
Our analysis proceeded in four steps. First, as a benchmark for our causal analysis, we assessed the "naïve" association between total years of schooling and our outcomes for cancer awareness and screening in the DHS 2009–10 and DHS 2014 samples. We assessed the crude bivariate relation graphically and then adjusted for covariates in descriptive multivariate ordinary least squares (OLS; linear probability) regression models. Second, we

determined the association between the school entry eligibility cut-off and the total years of schooling completed in our analytical sample. To do so, we assessed whether birth cohorts born after the June 30 cut-off had higher educational attainment than those who were born before the cut-off. We also estimated the relationship between exposure to the policy and schooling in multivariable OLS regression models (first stage). Third, we determined the relationship between our outcomes and women's month of birth (intention-to-treat). Fourth, not all women who were exposed to the school-entry age policy had higher educational attainment. We therefore scaled our intention-to-treat (ITT) results by the change in length of schooling resulting from exposure to the education policy (first stage), as shown in Eq. 1:

$$IV = \frac{ITT}{First\ stage} \tag{1}$$

To do so, we employed 2-stage least squares (2SLS) regression models using exposure to additional schooling from the June 30th cut-off as an instrumental variable (IV) for total years of schooling while adjusting for covariates (year of birth and survey)[24,25,41]. We assume that our IV estimates are local to the subpopulation who complied with their treatment assignment, meaning that these individuals experienced increased years of schooling due to the school eligibility cut-off. We provide additional details on the assumptions underpinning causal inference in Note S2 in the Appendix[41].

### Supplementary analyses

In addition to the analyses described above, we conducted several supplementary analyses to generate further confidence in our main results. First, causal inference using an exposure which is based on the month of birth could be undermined by seasonality in births or potential manipulation of birth dates. We, therefore, show the distribution in months of birth among young women and assess continuity in the density of birth dates across the June 30 threshold (Fig. S2). Second, we conducted a placebo test. To assess whether women in our analytical sample who were born before and after the cut-off were similar in predetermined characteristics, we plotted measured adult height by women's month of birth (Fig. S3). Adult height is largely determined by age five (prior to entering school) and should, therefore, not be affected by the school-entry policy in Lesotho. Third, to assess whether changes in educational attainment may explain the observed relationship between exposure to the school policy and our outcomes for cancer, we ran our main ITT regression models, described above, but additionally controlled for total years of schooling in these models. Our hypothesis was that if exposure to the education policy only affected cancer awareness and screening practices through changes in the duration of schooling, the ITT relationship is likely attenuated when controlling for schooling (Note S2). Fourth, breast cancer awareness was available among men in the DHS 2014. We, therefore, also assessed the relationship between education and breast cancer awareness among men (Note S3).

This was a complete case analysis using Stata MP v.17.

### Reporting summary

Further information on research design is available in the Nature Portfolio Reporting Summary linked to this article.

## Data availability

We used unrestricted data, which are publicly available upon request from the Demographic and Health Surveys Program (https://dhsprogram.com/). Dataset requests must include contact information, a research project title, and a description of the proposed analysis of the data.

## Code availability

All coding used in this study is available on the Harvard Dataverse repository. (https://doi.org/10.7910/DVN/ATGJYB).

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

## Acknowledgements

We thank the study participants and field staff of the Lesotho DHS 2009–10 and 2014. We are grateful to Jessica Cohen and Jacob Bor for their helpful comments on this project. J.L. and J.-W.D.N. were supported by the German Research Foundation (405898232). The funder had no role in study design, data collection and analysis, decision to publish, or preparation of the manuscript.

## Author contributions

J.L. and J.-W.D.N. conceived and designed the study, conducted the analyses, and wrote the first draft of the report. J.L., R.M., M.D.H., S.V.S., and J.-W.D.N. reviewed and contributed important revisions to the report. J.L. and J.-W.D.N. approved the final submitted version of the report. The corresponding author attests that all listed authors meet authorship criteria and that no others meeting the criteria have been omitted. J.L. and J.-W.D.N. had full access to all the data in the study and had final responsibility for the decision to submit for publication.

## Competing interests

The authors declare no competing interests.
