## [Peer Review file · Nature Communications]

Effect of girls' education on cancer awareness and screening in a natural experiment in Lesotho

Corresponding Author: Ms Janny Liao

Version 0:

Reviewer comments:

Reviewer #1

(Remarks to the Author)
Report

NCOMMS-24-35969, "The causal effect of girls' education on cancer awareness and screening: evidence from a natural experiment in Lesotho"

Short Synopsis

In many places, being born just before a cutoff date means starting school sooner. In Lesotho, being born in the first half of the year means starting school sooner. This in turn means being younger at any given grade. It turns out (and I hadn't previously known) that these younger-school-starters drop out of school more. In Lesotho, people who are born in July start school later, at an age almost a year older than people born in June; but by adulthood people born in July have half a year more schooling than those born in June. This natural experiment permits the estimation of causal effects of education on outcomes. This paper uses DHS data to do this.

This paper is focused on five outcomes. They are five survey questions:

1. Ever heard of breast cancer.
2. Ever done breast self exam.
3. Ever had breast clinical exam done.
4. Ever heard of pap smear.
5. Ever had pap smear done.

There appear to be small but positive effects of schooling for the two "heard of" questions, and for the last "had a pap smear" question, but neither of the breast exam questions.

Discussion

1. The identification strategy is interesting and pretty persuasive.
2. It would be helpful to see the robustness (Table S1) not just for one outcome, but also for the other four.
3. The greatest worry I have is that, as a respondent, saying you have heard of something or saying you have gotten a medical screening that is rare in the population (ten percent of control group women say they have had a pap smear) are both the kind of thing that social desirability bias could lead you to say. But unlike some other outcomes that many others in your community may know or which are easy to verify (how far you went in school, or what your home's floor is made of, for example), these are questions that are not easy to double check. I don't know what corroborating evidence can be brought to bear that these effects are not just effects of social desirability bias coming from those with more schooling. They could indeed be real.
4. For better interpretation and understanding, this is a less central point, but is there a particular level of schooling that July

birthdays are more likely than June birthdays to attain? Is there anything else written about why this dropout/attainment pattern is present?

5. I could not tell from my read, but I may have missed it: what other paper has looked at the causal effect of education on ever having heard of pap smears or breast cancer – is it the cited paper in the BMC Public Health?

Reviewer #2

(Remarks to the Author)

The authors find that increased education increases breast and cervical cancer awareness and the take up of screening measures. While education has been shown to causally affect many different health behaviors of women, this does seem to be the first evidence showing this particular relationship. Although it does mirror the existing literature for a host of other health outcome measures. The novelty in the paper really lies in their approach, where the authors use a creative application of the finding that the school age of entry policy in Lesotho led women to complete more schooling simply due to the month in which they were born. Thus, offering up a useful instrument for assessing the causal effect of more education on various breast and cervical cancer awareness and preventative measures.

In general, I found the exposition quite clear and compelling.

Some comments:

- It would be useful to interpret the effect sizes in context of other cervical or breast cancer interventions. For example, those included in this systematic review: <https://link.springer.com/article/10.1186/s12905-023-02265-8>. I think this would help the reader understand the broader significance of this as a policy tool relative to other tools available.
- I found the placebo result quite convincing. As a further check: If the sample is restricted to women born within a narrower window (say 3 months) of the cut-off does one find the same result? Put differently is this really driven by girls born around the cutoff. One will lose statistical precision, but if it is qualitatively similar that is helpful as further evidence in support of the findings in the supplemental material.
- The first sentence of the abstract: “Breast and Cervical cancers are leading causes of death in women in Sub-Saharan Africa.” Is at best misleading. The GBD report that is subsequently used to substantiate a similar but distinct statement in the introduction itself finds that among women 15-49 in sub-Saharan Africa, 1.1% and 1.37% of deaths are attributable to breast and cervical cancer; while among women 50-69, this is 2.42% and 2.38%. While deaths associated with breast cancer is increasing, the rate of change of deaths associated with cervical cancer is in fact declining. Clearly, these are important and consequential health risks for women, but a more nuanced framing of the issue seems appropriate.
- Line 215, I think this should refer to Figure S3
- Line 218, it would be useful to reference Figure S4
- Notes to Table 1: It was not 100% clear how missing values were treated in the construction of many of the awareness or screening variables. This table note states that the sample is restricted to women in the 25-49 age window with data on the breast and cervical awareness and screening questions. Is it correct to interpret this as if any question had missing information for one of these questions the women was dropped? Looking at the sample sizes in the DHS data this seems likely, but a bit more precision would be helpful for replication in the text.
- Several of the supplementary figures or tables are not cited within the text. I think many are helpful and would include links to all of the supplemental material.

Reviewer #3

(Remarks to the Author)

Thank you for the asking me to review this interesting manuscript on the causal effect of girls' education on cancer awareness: evidence from a natural experiment in Lesotho. This work presents a significant contribution to the field, particularly in its methodological approach and focus on an understudied context. The data analysis and interpretation were presented well.

The work generally supports its main claims and conclusions well, particularly regarding the causal effect of education on cancer awareness and screening in Lesotho. However, some broader claims about overall health impacts and generalizability to other contexts could benefit from additional evidence. The authors could consider tempering these broader claims or explicitly noting them as areas for future research.

An important comment is that the authors should add more explanation on natural experimentation, how it can substitute randomised control trial and how the school-entry age cut-off policy was used in the natural experiment.

Version 1:

Reviewer comments:

Reviewer #1

(Remarks to the Author)

Report on revised manuscript NCOMMS-24-35969A, “The causal effect of girls’ education on cancer awareness and

screening: evidence from a natural experiment in Lesotho”

I appreciate the inclusion of the new robustness tables. Some results may be seen as somewhat tenuous given that they do not all remain as statistically strong when restricting attention to the +/- 3 month window. Nonetheless the tables are very helpful.

I think the schooling outcome alone is interesting.

Three suggestions are provided below, all of which are easy to implement.

Two suggested revisions relating to the way literature is discussed.

1. There is a new sentence reading “In the United States, for example, a natural experiment investigated the effect of the Medicaid insurance program on health services utilization using exogenous variation in program enrollment produced when the Oregon state legislature assigned citizens to program eligibility using a random lottery.” I would suggest changing the wording so that “natural experiment” is replaced with “randomized trial.” The paper that is cited describes previous quasi-experimental work on health insurance setting the stage for the randomized experiment in Oregon. I would not call it a natural experiment when researchers intentionally randomize for the purpose of measuring treatment effects. I would call that a randomized trial.

2. The identification strategy in the present manuscript is very nice. I suggest that at some point in the paper adding a reference to a much earlier work leveraging variation in dates of birth vis-à-vis schooling laws. I think the quintessential reference would be: Angrist, Joshua D., and Alan B. Krueger. "Does compulsory school attendance affect schooling and earnings?." *The Quarterly Journal of Economics* 106, no. 4 (1991): 979-1014.

One suggested revision relating to limitations.

The paper has the new sentence: “Third, respondents may be influenced by social desirability bias when answering questions related to medical screening.” This is a very important caveat and I think the paper is made better by its inclusion. I suggest adding one other sentence after that one and before the next: “The outcomes in this study are survey-based; no official medical records or biomarkers are included as part of this study.”

Reviewer #2

(Remarks to the Author)

I have no additional comments, the authors have responded comprehensively to my comments.

Reviewer #1

In many places, being born just before a cutoff date means starting school sooner. In Lesotho, being born in the first half of the year means starting school sooner. This in turn means being younger at any given grade. It turns out (and I hadn't previously known) that these younger-school-starters drop out of school more. In Lesotho, people who are born in July start school later, at an age almost a year older than people born in June; but by adulthood people born in July have half a year more schooling than those born in June. This natural experiment permits the estimation of causal effects of education on outcomes. This paper uses DHS data to do this.

This paper is focused on five outcomes. They are five survey questions:

- 1. Ever heard of breast cancer.*
- 2. Ever done breast self exam.*
- 3. Ever had breast clinical exam done.*
- 4. Ever heard of pap smear.*
- 5. Ever had pap smear done.*

There appear to be small but positive effects of schooling for the two "heard of" questions, and for the last "had a pap smear" question, but neither of the breast exam questions.

- 1. The identification strategy is interesting and pretty persuasive.*

Authors' response:

We thank the Reviewer for these comments.

- 2. It would be helpful to see the robustness (Table S1) not just for one outcome, but also for the other four.*

Authors' response:

We have now added the results of supplementary robustness analyses for the other four outcomes, as recommended by the Reviewer (Tables S2-S6). Our results under different model and sample specifications are largely consistent with our main results (Table 4).

"Table S2. Sensitivity analyses: 2SLS regression results for breast cancer awareness among women when using alternative model and sample specifications" (Appendix, revised manuscript)

"Table S3. Sensitivity analyses: 2SLS regression results for breast self-exam among women when using alternative model and sample specifications" (Appendix, revised manuscript)

“Table S4. Sensitivity analyses: 2SLS regression results among women for breast clinical screening when using alternative model and sample specifications” (Appendix, revised manuscript)

“Table S5. Sensitivity analyses: 2SLS regression results among women for cervical cancer awareness when using alternative model and sample specifications” (Appendix, revised manuscript)

“Table S6. Sensitivity analyses: 2SLS regression results among women for cervical cancer screening when using alternative model and sample specifications” (Appendix, revised manuscript)

3. The greatest worry I have is that, as a respondent, saying you have heard of something or saying you have gotten a medical screening that is rare in the population (ten percent of control group women say they have had a pap smear) are both the kind of thing that social desirability bias could lead you to say. But unlike some other outcomes that many others in your community may know or which are easy to verify (how far you went in school, or what your home's floor is made of, for example), these are questions that are not easy to double check. I don't know what corroborating evidence can be brought to bear that these effects are not just effects of social desirability bias coming from those with more schooling. They could indeed be real.

Authors' response:

We thank the Reviewer for this excellent suggestion. We have now added results for several outcomes which were measured and/or more easily verified, as recommended by the Reviewer. Specifically, we have added results for measured literacy which was directly assessed and recorded by field interviewers, household wealth index, health insurance coverage, and travel time from the household to the nearest clinic (2 hours or less). The DHS-provided wealth index is a composite measure of a household's cumulative living standard, categorizing households into quintiles (1=poorest, 5=richest). The index is calculated using data on a household's ownership of assets, such as televisions, bicycles, and cars; dwelling characteristics, flooring material; type of drinking water source; and toilet and sanitation facilities.

In Figure S5, we graphically display the relationship between these outcomes and month of birth among women in Lesotho. In Table S10, we show ITT and 2SLS regression results controlling for birth cohort and period effects. Women who were born after the cutoff were on average considerably more literate, read more, lived in wealthier households, more often enrolled in health insurance, and lived in households which were located closer to a health facility. For example, each additional year of schooling as a result of the school-entry age policy increased measured literacy, health insurance coverage, and access to a health facility by 8.9 (s.e. = 0.013), 2.8 (s.e.= 0.010) and 8.5 (s.e. = 0.017) percentage points, respectively. These improvements in cognitive and economic capabilities among higher educated women likely contributed to increased cancer awareness and screening practices. Increased demand for

health insurance and access to healthcare facilities reflect impacts on health-seeking behavior and healthcare access, consistent with patterns observed for cancer-related outcomes.

“Figure S5. Intention-to-treat results: measured literacy, wealth and access to care by month of birth among women aged 25–49 years in the Lesotho DHS 2009-10 and 2014”

Notes: Figure shows measured literacy, household wealth, insurance coverage, and distance to care by month of birth among women aged 25–49 years with 95% confidence intervals. Women born in July-December had on average higher measured literacy (a), lived in wealthier households (b), were more likely to be enrolled in health insurance (c), and lived closer to a health facility (d). Literacy was defined by the DHS as a binary indicator which equals one if the respondent could read a whole or part of a sentence or attended secondary school or higher; and zero otherwise. Distance to health facility was defined as a binary indicator which equals one if the respondent lived within 2 hours travel time of a health facility; and zero otherwise. Unweighted. Source: data from Lesotho DHS 2009-10 and 2014. $N=7,971$.

“Table S10. ITT and 2SLS regression results: the relationship of girls’ education with measured cognitive skills, household wealth quintile, and access to healthcare”

Dependent variable (DV)	Measured literacy (1=yes, 0=no)	Reads news or magazine (1=yes, 0=no)	Household wealth quintile (1=poorest, 5=richest)	Enrolled in health insurance (1=yes, 0=no)	2 hrs or less to health facility (1=yes, 0=no)
	β	β	β	β	β
Model 1: ITT models controlling for period and birth cohort fixed effects					
Predictor: born between July and December (1=yes, 0=no)	0.050***	0.039***	0.245***	0.012*	0.048***
	(0.008)	(0.011)	(0.033)	(0.006)	(0.009)
R-squared	0.016	0.008	0.013	0.046	0.010
Observations	7,860	7,860	7,860	7,860	7,860
Model 2: 2SLS models controlling for period and birth cohort fixed effects					
Coefficient on years of schooling	0.089***	0.068***	0.432***	0.020**	0.085***
	(0.013)	(0.017)	(0.055)	(0.010)	(0.017)
Mean DV, January-June birth cohorts	0.841	0.306	3.06	0.076	0.774
F-statistic	62.6	62.6	62.6	62.6	62.6
Observations	7,860	7,860	7,860	7,860	7,860

Notes: Table shows intention-to-treat (ITT) and two-stage least squares regression results (2SLS) for the effect of girls’ education on women’s cognitive skills, household wealth, and access to healthcare. Model 1 is an ordinary least squares (OLS) linear probability model controlling for potential confounders. Model 2 is a 2-stage least squares linear probability model in which exposure to increased schooling from the school-entry age policy was used as an instrumental variable for the respondent’s duration of schooling (in years). All models controlled for indicators for year of birth and survey year. Robust standard errors in parentheses. The sample included all women aged between 25-49 years in the Lesotho DHS 2009-10 and 2014. The DHS-provided wealth index is a composite measure of a household’s cumulative living standard, categorizing households into quintiles (1=poorest, 5=richest). *** p<0.01, ** p<0.05, * p<0.1

4. For better interpretation and understanding, this is a less central point, but is there a particular level of schooling that July birthdays are more likely than June birthdays to attain? Is there anything else written about why this dropout/attainment pattern is present?

We have now added results for each level (year) of schooling, as recommended by the Reviewer. To do so, we determined the relationship between being born after the cutoff and a binary indicator for having completed at least [X] years of schooling, using separate models for having completed at least 1, 2, 3, ... , and 12 years of schooling or more. These results suggest that being born in July-December was associated with increased schooling at all levels (years), consistent with our main results. For example, women born between July and December were 6.5 percentage points and 4.6 percentage points more likely to have completed at least 6 years of schooling (primary or higher) or at least 10 years of schooling (junior secondary or higher), respectively, compared to women born between January and June (Table S1).

No paper, to our knowledge, has been published on the impacts of school-entry age on schooling and health outcomes in Africa. There could be several mechanisms which may explain these patterns in educational attainment. One hypothesis relates to differences in skill acquisition (human capital accumulation) between younger and older students in the same grade. Older students may be more cognitively and non-cognitively mature than their younger classmates despite no differences in innate ability, consistent with our results for improvements in measured literacy skills and reading (Figure S5 and Table S10). Starting school later has also been linked to better test performance in school in other settings (Peña, 2017).

These differences in human capital trajectories by school-entry age may be further accentuated in the context of Lesotho where class sizes are large, there is considerable heterogeneity in skills in school, and the opportunity costs of schooling may be greater compared to those in higher income settings. For example, school-aged children frequently work on family farms, herd livestock, provide child and elder care, perform other home production activities, and enter the labor force, suggesting strong opportunity costs to schooling. Households and teachers may also invest more in old-for-grade children based on their perceived school performance (rather than actual performance), ultimately leading to real differences in human capital acquisition.

Reference:

Peña PA (2017). Creating winners and losers: Date of birth, relative age in school, and outcomes in childhood and adulthood. *Economics of Education Review*, 56, 152–176.

“Table S1. Intention-to-treat regression results for the relationship between being born after June 30th and year of schooling completed among women”

Predictor: born between July and December (1=yes, 0=no)	β	Mean DV, January-June birth cohorts	R-squared	N
Dependent variable (DV)				
1: Has completed at least 1 year of schooling (1=yes, 0=no)	0.013*** (0.003)	0.971	0.009	7,971
2: Has completed at least 2 years of schooling (1=yes, 0=no)	0.016*** (0.004)	0.965	0.012	7,971
3: Has completed at least 3 years of schooling (1=yes, 0=no)	0.020*** (0.004)	0.952	0.012	7,971
4: Has completed at least 4 years of schooling (1=yes, 0=no)	0.032*** (0.005)	0.926	0.018	7,971
5: Has completed at least 5 years of schooling (1=yes, 0=no)	0.046*** (0.006)	0.879	0.025	7,971
6: Has completed at least 6 years of schooling (1=yes, 0=no)	0.065*** (0.008)	0.805	0.026	7,971
7: Has completed at least 7 years of schooling (1=yes, 0=no)	0.084*** (0.010)	0.668	0.033	7,971
8: Has completed at least 8 years of schooling (1=yes, 0=no)	0.087*** (0.011)	0.367	0.031	7,971
9: Has completed at least 9 years of schooling (1=yes, 0=no)	0.077*** (0.011)	0.291	0.026	7,971
10: Has completed at least 10 years of schooling (1=yes, 0=no)	0.046*** (0.009)	0.208	0.022	7,971
11: Has completed at least 11 years of schooling (1=yes, 0=no)	0.033*** (0.009)	0.161	0.021	7,971
12: Has completed at least 12 years of schooling (1=yes, 0=no)	0.023*** (0.007)	0.113	0.012	7,971

Notes: Table shows ordinary least regression (OLS) results for the relationship between being born after June 30th (binary) and an indicator for having completed at least X years of schooling, estimated separately for having completed at least 1, 2, ... , and 12 years of schooling. All models controlled for year of birth and survey year. Robust standard errors in parentheses. The sample includes all women ages 25-49 years in the Lesotho DHS 2009-10 and 2014 (N=7,971). *** p<0.01, ** p<0.05, * p<0.1

5. I could not tell from my read, but I may have missed it: what other paper has looked at the causal effect of education on ever having heard of pap smears or breast cancer – is it the cited paper in the *BMC Public Health*?

Authors' response:

No other study, to our knowledge, has looked at the causal effect of education on breast cancer and pap smear screening practices in the context of low- and middle-income countries. There are several observational studies from Lesotho (Afaya et al. 2023), Cameroon (Okyere et al, 2021), and India (Sen et al, 2022), but these studies are vulnerable to residual bias. A study from Sweden has looked at the effect of education on breast cancer incidence by exploiting an education reform that extended compulsory schooling (Palme and Simeonova 2015). However, the study is limited to breast cancer and is difficult to generalize to Lesotho.

References:

Afaya A, Japiong M, Konlan KD, et al. Factors associated with awareness of breast cancer among women of reproductive age in Lesotho: a national population-based cross-sectional survey. *BMC Public Health*. 2023 Mar 31;23(1):621.

Okyere J, Duodu PA, Aduse-Poku L, et al. Cervical cancer screening prevalence and its correlates in Cameroon: secondary data analysis of the 2018 demographic and health surveys. *BMC Public Health*. 2021 Jun 5;21(1):1071.

Sen S, Khan PK, Wadasadawala T, et al. Socio-economic and regional variation in breast and cervical cancer screening among Indian women of reproductive age: a study from National Family Health Survey, 2019-21. *BMC Cancer*. 2022 Dec 7;22(1):1279.

Palme M, et al. Does women's education affect breast cancer risk and survival? Evidence from a population based social experiment in education. *J Health Econ*. 2015 Jul;42:115-24.

Reviewer #2

The authors find that increased education increases breast and cervical cancer awareness and the take up of screening measures. While education has been shown to causally affect many different health behaviors of women, this does seem to be the first evidence showing this particular relationship. Although it does mirror the existing literature for a host of other health outcome measures. The novelty in the paper really lies in their approach, where the authors use a creative application of the finding that the school age of entry policy in Lesotho led women to complete more schooling simply due to the month in which they were born. Thus, offering up a useful instrument for assessing the causal effect of more education on various breast and cervical cancer awareness and preventative measures.

In general, I found the exposition quite clear and compelling.

Authors' response:

We thank the Reviewer for these comments.

Some comments: It would be useful to interpret the effect sizes in context of other cervical or breast cancer interventions. For example, those included in this systematic review: <https://link.springer.com/article/10.1186/s12905-023-02265-8>. I think this would help the reader understand the broader significance of this as a policy tool relative to other tools available.

Authors' response:

We thank the Reviewer for bringing this helpful point to our attention. We have now provided additional discussion on effect sizes in the context of other interventions. As a supplementary analysis, we have now also provided adjusted relative risk ratios (RR) to further facilitate comparisons across different interventions and settings (Table S7). In terms of relative effect size, our results are comparable to other known interventions (Tin et al., 2023). In Nigeria, for example, a 6-month mHealth intervention with health promotion information increased the uptake of cervical cancer screening by 43% compared to standard of care (RR: 1.43; 95% CI: 1.03-1.98). Similarly, in our study, women with an additional year of schooling as a result of the school-entry age policy in Lesotho were 41% more likely to have been screened for cervical cancer (RR: 1.41; 95% CI: 1.06-1.88) (Table S7). Importantly, however, formal education has a myriad of additional benefits which are not taken into account in our analyses.

“Table S7. Sensitivity analyses: using Poisson regression models to estimate adjusted relative risk ratios for the effect of education on cancer awareness and screening”

Dependent variable (DV)	Ever heard of breast cancer (1=yes, 0=no)	Breast self exam (1=yes, 0=no)	Breast clinical exam (1=yes, 0=no)	Ever heard of Pap smear (1=yes, 0=no)	Performed Pap smear (1=yes, 0=no)
	RR	RR	RR	RR	RR
Coefficient on years of schooling	1.057**	1.045	0.991	1.182***	1.408**
	(1.011-1.105)	(0.923-1.183)	(0.739-1.330)	(1.070-1.306)	(1.056-1.878)
Mean DV, January-June birth cohorts	0.877	0.320	0.078	0.444	0.100
Observations	3,763	7,971	7,971	7,971	7,971

Notes: Adjusted relative risk ratios (RR) from multivariable Poisson regression models with 95% with confidence intervals (IVpoisson). All models control for year of birth and survey year. Exposure to increased schooling from the school-entry age policy was used as an instrumental variable for the respondent's duration of schooling (in years). The sample included all women aged between 25-49 years in the Lesotho DHS 2009-10 and 2014. Data on breast cancer awareness was not available in the DHS 2009-10 survey. *** p<0.01, ** p<0.05, * p<0.1.

References:

Tin KN, Ngamjarus C, Rattanakanokchai S et al. Interventions to increase the uptake of cervical cancer screening in low- and middle-income countries: a systematic review and meta-analysis. *BMC Women's Health* 23, 120 (2023).

Okunade KS, Soibi-Harry A, John-Olabode S, et al. Impact of Mobile Technologies on Cervical Cancer Screening Practices in Lagos, Nigeria (mHealth-Cervix): A Randomized Controlled Trial. *JCO Glob Oncol.* 2021 Aug;7:1418-1425.

I found the placebo result quite convincing. As a further check: If the sample is restricted to women born within a narrower window (say 3 months) of the cut-off does one find the same

result? Put differently, is this really driven by girls born around the cutoff. One will lose statistical precision, but if it is qualitatively similar that is helpful as further evidence in support of the findings in the supplemental material.

Authors' response:

We thank the Reviewer for this suggestion. We have now added a supplementary analysis where we restricted the sample to women born within a narrower window around the school-entry age cutoff in Lesotho, as recommended by the Reviewer. In addition to a window of +/- 6 months around the cutoff (figures a and b), we now show a window of +/- 3 months around the cutoff (figures c and d) with qualitatively similar results (Figure S3).

“Figure S3. Placebo outcome: measured adult height (in cm) by month of birth among women aged 25–49 years in the Lesotho DHS 2009-10 and 2014”

Notes: Figure shows measured adult height (in cm) by month of birth among women aged 25–49 years with 95% confidence intervals, separately by survey year. Unweighted. The sample in figures (a) and (b) includes all women born between January and December ($N=7,971$). The sample in figures (c) and (d) includes all women born between April and September ($N=3,988$). Source: data from Lesotho Demographic and Health Surveys (DHS) 2009-10 and 2014. $N=7,971$.

The first sentence of the abstract: “Breast and Cervical cancers are leading causes of death in women in Sub-Saharan Africa.” is at best misleading. The GBD report that is subsequently used to substantiate a similar but distinct statement in the introduction itself finds that among women 15-49 in sub-Saharan Africa, 1.1% and 1.37% of deaths are attributable to breast and cervical cancer; while among women 50-69, this is 2.42% and 2.38%. While deaths associated with breast cancer are increasing, the rate of change of deaths associated with cervical cancer is in fact declining. Clearly, these are important and consequential health risks for women, but a more nuanced framing of the issue seems appropriate.

Authors’ response:

We thank the Reviewer for bringing this to our attention and have rephrased the sentence. We would be happy to make additional changes at the behest of the Reviewers and/or Editors.

“Breast and cervical cancers are important causes of cancer-related disability and premature death for women in Sub-Saharan Africa.” (Abstract, revised manuscript)

Line 215, I think this should refer to Figure S3

Authors’ response:

Thank you for bringing this to our attention. We have now corrected this in the revised manuscript (former Figure S3 has become Figure S2 during revisions).

Line 218, it would be useful to reference Figure S4

Authors’ response:

We have now added a reference to the figure (now Figure S3).

Notes to Table 1: It was not 100% clear how missing values were treated in the construction of many of the awareness or screening variables. This table note states that the sample is restricted to women in the 25-49 age window with data on the breast and cervical awareness and screening questions. Is it correct to interpret this as if any question had missing information for one of these questions the women was dropped? Looking at the sample sizes in the DHS data this seems likely, but a bit more precision would be helpful for replication in the text.

Authors’ response:

The Reviewer is correct, and we have now further clarified this throughout the paper. Among women who completed the interview, fewer than 1% did not complete all breast and cervical cancer awareness and screening questions. We also show a study participant flow diagram (Figure S1) in the Appendix which shows how the final analytical sample was derived.

“Women with missing information for any of the five breast and cervical cancer awareness and screening questions were removed from the analytical sample (n=36), yielding a final analytical sample of 7,971 women.” (Table 1, revised manuscript)

“This was a complete case analysis.” (p.11, revised manuscript)

“Figure S1. Study participant flow diagram” (Appendix, revised manuscript)

Several of the supplementary figures or tables are not cited within the text. I think many are helpful and would include links to all of the supplemental material.

Authors' response:

We thank the Reviewer for this comment. We have now provided additional citations to the supplementary material and cite all supplementary figures and tables.

Reviewer #3

Thank you for asking me to review this interesting manuscript on the causal effect of girls' education on cancer awareness: evidence from a natural experiment in Lesotho. This work presents a significant contribution to the field, particularly in its methodological approach and focus on an understudied context. The data analysis and interpretation were presented well.

Authors' response:

We thank the Reviewer for these comments.

The work generally supports its main claims and conclusions well, particularly regarding the causal effect of education on cancer awareness and screening in Lesotho. However, some broader claims about overall health impacts and generalizability to other contexts could benefit

from additional evidence. The authors could consider tempering these broader claims or explicitly noting them as areas for future research.

Authors' response:

We thank the Reviewer for this helpful suggestion. We have now rephrased broader claims about overall health impacts throughout the paper, noted the issue of external generalizability to other outcomes and contexts in the Discussion section, and more explicitly noted future directions for research in the Conclusion section. We have also provided additional supporting evidence on impacts which may be relevant for other health outcomes. Specifically, we find that women with more educational attainment as a result of the school-entry age policy were more likely to be literate, lived in wealthier households, were more likely to be enrolled in health insurance, and had better access to health facilities (Figure S5 and Table S10).

“Our study may not generalize to other settings.” (p.27, revised manuscript)

“Second, while our data includes participants up to the age of 45 years old, they may not be generalizable to older age groups. Older age groups could exhibit different educational impacts on health behaviors.” (p.27, revised manuscript)

“Additional empirical evidence is needed to determine whether improvements in educational attainment translate to reduced disability and premature deaths, lead to similar outcomes in the awareness and screening of other types of cancers, such as colorectal or lung cancer, and extend to broader health domains, such as chronic disease management and preventive care practices. Future studies could build on this natural experiment to better understand the full scope of impact of girls' education on women's health in Lesotho.” (p.28, revised manuscript)

“Table S10. ITT and 2SLS regression results: the relationship of girls' education with measured cognitive skills, household wealth, and access to healthcare”

“Figure S5. Intention-to-treat results: measured literacy, wealth and access to care by month of birth among women aged 25–49 years in the Lesotho DHS 2009-10 and 2014”

Notes: Figure shows measured literacy, household wealth, insurance coverage, and distance to care by month of birth among women aged 25–49 years with 95% confidence intervals. Women born in July-December had on average higher measured literacy (a), lived in wealthier households (b), were more likely to be enrolled in health insurance (c), and lived closer to a health facility (d). Literacy was defined by the DHS as a binary indicator which equals one if the respondent could read a whole or part of a sentence or attended secondary school or higher; and zero otherwise. Distance to health facility was defined as a binary indicator which equals one if the respondent lived within 2 hours travel time of a health facility; and zero otherwise. Unweighted. Source: data from Lesotho DHS 2009-10 and 2014. $N=7,971$.

An important comment is that the authors should add more explanation on natural experimentation, how it can substitute randomised control trial and how the school-entry age cut-off policy was used in the natural experiment.

Authors' response:

We agree with the Reviewer and have now provided more discussion on the value of natural experiments and key advantages throughout the revised manuscript. Additionally, we have provided more details on the education context in Lesotho (Text S1), including the school-entry age policy and its effects on the distribution of educational attainment (Table S1). We would be happy to make additional changes, however, at the behest of the Reviewer and/or Editor.

“In the absence of randomized controlled trials (RCTs), natural experiments offer an important opportunity for causal inference in population health.^{17,18} In the United States, for example, a natural experiment investigated the effect of the Medicaid insurance program on health services utilization using exogenous variation in program enrollment produced when the Oregon state legislature assigned citizens to program eligibility using a random lottery.¹⁹ Natural experiments can complement evidence from RCTs and have several strengths.²⁰ For example, to evaluate interventions that have become standard practice without preceding RCTs or to estimate a policy's “real-life” effectiveness as opposed to clinical efficacy.²¹ Policy makers, for example, may be more concerned with *de facto* effects of large-scale programs as opposed to effects under ideal conditions such as during an RCT.²² Additionally, natural experiments can be less costly because they can be implemented using existing data. Moreover, in studies focused on formal education, it would be ethically impermissible to randomly allocate individuals to receive or be deprived of educational opportunities.” (p.4-5, revised manuscript).

“The education system in Lesotho is organized into basic education (grades 1 to 10) and secondary education (grades 11 and 12) (Raselimo and Mahao, 2015). Primary school education has been free since 2000, and attendance is high. However, fees are assessed to attend secondary school. Despite the high transition rate from primary to secondary school, many children in Lesotho do not complete secondary school, partly due to costs (UNESCO 2024). Per the Lesotho Education Act of 2010, “a parent shall enroll a learner in a primary school at the age of six years or in the year in which he or she will be six years of age by the 30th of June of that calendar year”. As a result of the school-entry age policy, July-born children are on average older compared to June-born children at the beginning of grade 1. Despite starting school at older ages, July-born have completed more total years of schooling than those born in June by late adolescence and early adulthood (De Neve, Moshoeshoe, Bor 2023).

There may be several explanations for the observed differences in total educational attainment in adulthood by month of birth in Lesotho. For example, late school entrants are old-for-grade and may be more cognitively and non-cognitively mature compared to their younger classmates despite no differences in innate ability. Because of these

differences in maturity and/or self-confidence in school, late entrants into primary school may accumulate more human capital (Peña PA 2017). These differences in human capital trajectories by month of birth may be further accentuated in the context of Lesotho where class sizes are large, there is considerable heterogeneity in skills in school, and there are considerable opportunity costs to schooling. For example, school-age children frequently herd livestock, provide child and elder care, or perform other home production activities. Households and teachers may also invest more in old-for-grade children based on their perceived school performance (rather than actual performance), ultimately leading to real differences in human capital acquisition.” (Appendix, revised manuscript).

Reviewer #1

Report on revised manuscript NCOMMS-24-35969A, "The causal effect of girls' education on cancer awareness and screening: evidence from a natural experiment in Lesotho"

I appreciate the inclusion of the new robustness tables. Some results may be seen as somewhat tenuous given that they do not all remain as statistically strong when restricting attention to the +/- 3 month window. Nonetheless the tables are very helpful.

I think the schooling outcome alone is interesting.

Authors' response:

We thank the Reviewer for these comments.

Three suggestions are provided below, all of which are easy to implement. Two suggested revisions relating to the way literature is discussed.

1. There is a new sentence reading "In the United States, for example, a natural experiment investigated the effect of the Medicaid insurance program on health services utilization using exogenous variation in program enrollment produced when the Oregon state legislature assigned citizens to program eligibility using a random lottery." I would suggest changing the wording so that "natural experiment" is replaced with "randomized trial." The paper that is cited describes previous quasi-experimental work on health insurance setting the stage for the randomized experiment in Oregon. I would not call it a natural experiment when researchers intentionally randomize for the purpose of measuring treatment effects. I would call that a randomized trial.

Authors' response:

We thank the Reviewer for these comments. We have now further clarified the sentence, as recommended by the Reviewer. Specifically, to avoid potential confusion between approaches, we have now added a more directly related example of a natural experiment from the region.

2. *The identification strategy in the present manuscript is very nice. I suggest that at some point in the paper adding a reference to a much earlier work leveraging variation in dates of birth vis-à-vis schooling laws. I think the quintessential reference would be: Angrist, Joshua D., and Alan B. Krueger. "Does compulsory school attendance affect schooling and earnings?." The Quarterly Journal of Economics 106, no. 4 (1991): 979-1014.*

Authors' response:

We have now added the reference as recommended by the Reviewer.

One suggested revision relating to limitations. The paper has the new sentence: "Third, respondents may be influenced by social desirability bias when answering questions related to medical screening." This is a very important caveat and I think the paper is made better by its inclusion. I suggest adding one other sentence after that one and before the next: "The outcomes in this study are survey-based; no official medical records or biomarkers are included as part of this study."

Authors' response:

We have now added the sentence as recommended by the Reviewer.

Reviewer #2

I have no additional comments, the authors have responded comprehensively to my comments.

Authors' response:

We thank the Reviewer for these comments.